# First Report of Complete Mitochondrial Genome in the Tribes Coomaniellini and Dicercini (Coleoptera: Buprestidae) and Phylogenetic Implications

**DOI:** 10.3390/genes13061074

**Published:** 2022-06-16

**Authors:** Xuyan Huang, Bo Chen, Zhonghua Wei, Aimin Shi

**Affiliations:** College of Life Sciences, China West Normal University, Nanchong 637009, China; huangxy9999@gmail.com (X.H.); chenb9321@gmail.com (B.C.); wzh1164@126.com (Z.W.)

**Keywords:** Buprestidae, Coomaniellini, Dicercini, mitogenome, phylogenetic analysis

## Abstract

The complete mitochondrial genomes (mitogenomes) of the tribes Coomaniellini and Dicercini were sequenced and described in this study, including *Coomaniella copipes* (16,196 bp), *Coomaniella dentata* (16,179 bp), and *Dicerca corrugata* (16,276 bp). These complete mitogenomes are very similar in length and encoded 37 typical mitochondrial genes, including 22 transfer RNA genes (tRNAs), 2 ribosomal RNA genes (rRNAs) and 13 protein-coding genes (PCGs). Most of PCGs had typical ATN start codons and terminated with TAR. Among these mitogenomes, Leu2 (L2), Ile (I), Ser2 (S2), and Phe (F) were the four most frequently encoded amino acids. Moreover, phylogenetic analyses were performed based on three kinds of nucleotide matrixes (13 PCGs, 2 rRNAs, and 13 PCGs + 2 rRNAs) among the available sequenced species of the family Buprestidae using Bayesian inference and Maximum-likelihood methods. The results showed that a Chrysochroninae species interspersed in Buprestinae, and Coomaniellini is more closely related to Dicercini than Melanophilini. Moreover, the clade of Buprestidae was well separated from outgroups and the monophyly of Agrilinae is confirmed again. Our whole mitogenome phylogenetic results support that the genus *Dicerca* can be transferred from Chrysochroinae to Buprestinae; whether Dicercini can be completely transferred remains to be further verified after enriching samples. Our results have produced new complete mitogenomic data, which will provide information for future phylogenetic and taxonomic research.

## 1. Introduction

The family Buprestidae is one of the largest families in Coleoptera, which comprises six subfamilies, 521 genera, and more than 15,000 species [1,2], widely distributed in the world. Some are important forestry and agricultural pests that threaten forest ecosystems and damage economical crops, especially in the subfamilies Agrilinae [3,4,5,6,7] and Buprestinae [8,9].

Although some taxonomists have made various contributions to the Buprestidae classification based on morphological characteristics [1,2,10,11,12,13,14], the phylogenetic relationships in subfamilies and tribes are still unresolved completely, such as: subfamilies Buprestinae and Chrysochroinae, tribes Tracheini, Agrilini, Coraebini, and Melanophilini, etc. Moreover, there is very little work on molecular phylogeny of higher taxa. Evans et al. [15] made the first large-scale molecular phylogeny of Buprestidae, results showed that the subfamilies Chrysochroinae and Buprestinae were polyphyletic, while the Agrilinae, Julodinae, and Galbellinae were monophyletic; the tribes Agrilini, Coraebini, and Tracheini were polyphyletic. Recently, molecular species delimitation was applied to Buprestidae. This method was used to analyze the *Chrysobothris femorata* species group by Hansen et al. [16], the *Agrilus* species by Pentinsaari et al. [17], Pellegrino et al. [18], and Kelnarova et al. [19], their results showed that DNA barcoding is a powerful species identification tool.

Additionally, the phylogenetic relationships of Melanophilini, Coomaniellini, and Dicercini remain unsettled. The tribe Dicercini comprises 29 genera widely distributed in Asia, Europe, Africa, South America, and North America [1,2]. One of the authors found the *Dicerca* species inhabit in plants *Pinus* sp. the biological characteristics of which are less known. The monogeneric tribe Coomaniellini was proposed by Bílý [20]. The adults of *Coomaniella* were usually found on leaves of Anacardiaceae [21,22]. All the known species of *Coomaniella* are distributed in South and Southeast Asia. The tribe Melanophilini comprises seven genera: *Juniperella*, *Melanophila*, *Phaenops*, *Trachypteris*, *Xenomelanopha*, *Merimna*, and *Cromophila* [1,2], widely distributed in Africa, Europe, Asia, North America, and South America. To date, the position of these tribes had yet been resolved in taxonomical classification.

In the past two decades, mitogenome emerged as an important source on phylogenetic analysis [23,24,25,26,27], evolution strategies [28,29,30,31], and genetic diversity [32,33,34] and species delimitation [35]. In 2009, the first mitogenome in Buprestidae (*Chrysochroa fulgidissima*) was published [36]. Xiao et al. [37] reported the mitogenome of *Trachys auricollis* and carried out a phylogenetic analysis of Elateriformia, which showed that the Buprestoidea is a sister group to Byrrhoidea. Sun et al. [38] also performed a phylogenetic analysis of suborder Polyphaga, and the results showed that the Buprestidae is a sister group to other Polyphaga infraorders. To date, 18 complete mitogenome sequences have been reported (including three species in this study), details are shown in Table 1.

In this study, the mitogenome sequences of *Dicerca corrugata*, *Coomaniella copipes,* and *C. dentata* are determined and annotated, which are the first complete mitogenome sequences reported in the tribes Coomaniellini and Dicercini. And the mitogenome of *Melanophila acuminata* was reported by Peng et al. [39], which belongs to the tribe Melanophilini. A preliminary phylogenetic analysis was undertaken to validate the phylogenetic position of the tribes Melanophilini, Coomaniellini, and Dicercini based on the mitogenomes. The results of this study may provide new data for phylogenetic studies of the Buprestidae and expand our knowledge of the mitochondrial genomic features of Buprestidae and the taxonomy within the family Buprestidae.

## 2. Materials and Methods

### 2.1. Sampling and DNA Extraction

The specimens of *Dicerca corrugata* were collected from plant *Pinus kwangtungensis* in Rufeng, Nanling National Forest Park, Shaoguan City, Guangdong Province, China (24.898625° N, 113. 24.01945° E, alt. 1542 m) on 23 May 2021. The specimens of *Coomaniella dentata* and *C. copipes* were collected from Guitian Village, Dayao Mountains, Jinxiu County, Guangxi Zhuang Autonomous Region, China (24.07335° N, 110.16771° E, alt. 800–1000 m), on 20 April 2021. These specimens were preserved in 95% ethanol at −24 °C for the long-term storage in the specimen collection room at China West Normal University. The total genomic DNA was extracted from muscle tissue of individual specimens using the Ezup Column Animal Genomic DNA Purification Kit (Shanghai, China) following the manufacturer’s instructions. For sequencing, the extracted DNA was stored at −24 °C.

### 2.2. Sequence Assembly, Annotation, and Analysis

Next-generation sequencing and assembly were performed by Beijing Aoweisen Gene Technology Co. Ltd. (Beijing, China) to obtain the complete mitogenome sequences of the three Buprestidae species. A whole genome shotgun strategy was used based on the Illumina HiSeq platform when the total genome DNA was quantified. The sequencing with a strategy of 150 bp paired-end reads. The assembly method of Hahn et al. [46] was used. The complete mitogenome sequences of three Buprestidae species were annotated using Geneious 11.0.2 [47] based on the invertebrate genetic code. The 22 tRNA genes were re-verified using MITOS webserver [48] based on the invertebrate mitogenetic code. The tRNA secondary structures were predicted using tRNAscan-SE [49]. The mitogenome maps were drawn using Organellar Genome DRAW [50]. The base composition and relative synonymous codon usage (RSCU) values were calculated using MEGA 7.0 [51]. Strand asymmetry was calculated in terms of formulae: AT-skew = (A − T)/(A + T) and GC-skew = (G − C)/(G + C) [52]. Nucleotide diversity (Pi), non-synonymous substitutions (Ka), and synonymous substitutions (Ks) of 13 PCGs were calculated using DnaSP v 5 [53].

### 2.3. Phylogenetic Analysis

A total of 18 buprestid complete mitogenomes (Table 1), including three newly sequenced species were used to construct the phylogenetic tree of 18 species from 11 genera belonging to 4 subfamilies of Buprestidae, with *Heterocerus parallelus* (Heteroceridae) and *Dryops ernesti* (Dryopidae) as outgroups based on the phylogenetic analyses of Polyphaga [37]. Three datasets (13 PCGs, 2 rRNAs, and 13 PCGs + 2 rRNAs) were used to construct the phylogenetic trees using PhyloSuite v 1.2.2 [54] based on the Maximum likelihood (ML) and Bayesian inference (BI) methods using different best-fit substitution models. The sequences were aligned using ClustalW [55] and trimmed by trimAl v 1.2 [56]. The best-fit models used in ML and BI analyses were calculated with ModelFinder [57]. The phylogenetic trees were reconstructed using IQ-tree v 1.6.8 [58] and MrBayes v 3.2.6 program [59] based on the ML and BI methods, respectively. Maximum likelihood analyses were run with 1000 ultrafast bootstrap and 1000 SH-aLRT replicates to estimate node reliability. Bayesian analyses were run with two independent chains spanning 2,000,000 generations, four Markov chains, sampling at every 100 generations, and with a burn-in of 25%. The phylogenetic trees were visualized and edited using FigTree v1.4 [60].

## 3. Results and Discussion

### 3.1. Genome Organization and Base Composition

We sequenced and annotated the complete mitogenomes of the buprestid species *C. copipes* (GenBank accession no. OL694145; SRA accession no. SRR19612349; length: 16,196 bp), *C. dentata* (OL694144; SRR19612367; 16,179 bp), and *D. corrugata* (OL753086; SRR19629749; 16,276 bp). Overall, these three mitogenomes have the same composition, consisting of 37 coding genes (13 PCGs, 22 tRNA, and two rRNA) and a non-coding A + T-rich region. Four PCGs (*nad5*, *nad4*, *nad4l*, and *nad1*), eight tRNAs (*trnQ*, *trnC*, *trnY*, *trnF*, *trnH*, *trnP*, *trnL1*, and *trnV*), and two rRNAs are encoded on the N-strand, while the other 23 genes (9 PCGs, 14 tRNAs) and the A + T-rich region are encoded on the J-strand (Figure 1 and Appendix A).

The three complete mitogenomes had high A + T contents of 71.76–76.59% and had a positive AT-skews and negative GC-skews. The overall AT-skews in the three entire mitogenomes were 0.03, 0.01, and 0.09, respectively (Table 1). The total length of 13 overlapping regions in the whole mitogenome of *C. copipes* is 37 bp, while *C. dentata* has 10 overlapping regions with a length of 33 bp, and D. corrugata has 12 overlapping regions, the length is 38 bp (Table 2). However, the longest overlapping regions of all three species are 8 bp, located between *trnW* and *trnC*, which is consistent with the results of some other studies on Buprestidae species with only slight differences in length, such as the longest overlapping region of 9 bp for *Chrysochroa fulgidissima* [36] and 8 bp for *Trachys troglodytiformis* [37]. In these three species, there is a 7 bp overlapping region between *atp8* and *atp6*, *nad4* and *nad4l*, which is often reported in the mitochondrial genome of insects.

The gene arrangement, nucleotide composition, and codon usage of these three mitogenomes were consistent with other buprestid species [36,37,38,39,40,42,43,44]. The same order of genes is not enough to unify taxa, but genes with the same rearrangement can be an effective marker of common ancestor [61]. The arrangement and rearrangement of genes may be used as a very favorable means to assist classification and infer ancient evolutionary relationships, since rearrangement is usually a unique and rare event [62]. The rearrangement phenomenon has been found in some studies [63,64,65,66], but it has not been detected in these three species.

### 3.2. Protein-Coding Regions, Codon Usage and Nucleotide Diversity

In these three mitogenomes, the regions of PCGs were 11,159 bp (*C. copipes*), 11,159 bp (*C. dentata*) and 11,150 bp (*D. corrugata*) in size, accounting for 75.41–78.66% of the entire mitogenome. The mitogenomes can be converted into 3707–3710 amino acid-coding codons, excluding stop codons (29 bp). We found the *atp8* and *nad5* to be the smallest (156 bp) and the largest (1720 bp) genes, respectively, which is similar to the other buprestid mitogenomes. The majority of PCGs directly use ATN (ATA/ATT/ATG/ATC) as the start codon, the exception is *nad1* gene, which starts with TTG. The unusual start codon of the *nad1* gene can be found in the mitogenomes of some other insects, such as *Trachys auricollis* (TTG) and *Liriomyza trifolii* (GTG) [38,67]. The start codon of *cox1* gene has not been determined in this study; nevertheless, it can be found in some reports that *cox1* is always nonstandard, for example, CGA in *Laelia suffuse* [64], AAA in *Tribolium castaneum* [68], or even ATCA in *Liriomyza trifolii* [67]. Moreover, except for five PCGs (*cox1*, *cox2*, *cox3*, *nad5,* and *nad4*) with an incomplete stop codon T-, the other eight PCGs have complete stop codons (TAA/TAG). The stop codon T- of the five genes was completed by the addition of 3′ A residues to the mRNA [69,70]. This phenomenon of generating TAA ends by post-transcriptional polyadenylation has been seen in many studies [36,37,65].

A summary of the number of amino acids in annotated PCGs (Figure 2A) is presented, along with the percentage of the top 11 amino acids with higher numbers (Figure 2B) and relative synonymous codon usage (Figure 3; Appendix A). We can conclude that the overall codon usage among the newly sequenced species was similar, with L2, I, S2, and F being the four most frequently used amino acids, and TTA (L2), ATT (I), TTT (F), and ATA (M) are the most frequently used codons.

The nucleotide diversity (Pi) of the 13 PCGs among three newly sequenced species is provided (Figure 4), which ranged from 0.159 to 0.297. In these genes, *atp8* (Pi = 0.297) presented the highest variability, followed by *nad6* (Pi = 0.292), *nad2* (Pi = 0.264), and *nad3* (Pi = 0.261); *cox1* (Pi = 0.159) exhibited the lowest variability. For the study of insect taxonomy and the analysis of species evolution, the mitochondrial gene *cox1* was primarily used because it is relatively conservative (Appendix A). The ratio of Ka/Ks for each gene of the 13 PCGs was calculated (Figure 5). The value of *nad1* gene is obviously higher than others, which indicates that the *nad1* gene has a relatively higher evolutionary rate. Meanwhile, the Ka/Ks ratios of the other 12 PCGs were all significantly less than 1, and the value of the *cox1* gene is the lowest (Ka/Ks = 0.05). Indeed, *cox1* shows the lowest Ka/Ks ratio in almost all insects, that is, a relatively low variation rate. Moreover, Xiao et al. [37] indicated that not only insects but almost all animals have the lowest Ka/Ks ratio for *cox1*. This indicates that the gene was subjected to the highest purifying selection [71]. The genes with the lowest and highest Ka were *cox1* (0.044) and *atp8* (0.286), respectively, while the Ks were *nad1* (0.189) and *nad3* (1.553) were the lowest and highest genes.

### 3.3. Ribosomal and Transfer RNA Genes

The lengths of *rrnL* genes ranged from 1282 bp (*D. corrugata*) to 1289 bp (*C. copipes*), whereas those of *rrnS* ranged from 713 bp (*C. dentata*) to 739 bp (*D. corrugata*). The AT content of the rRNA genes ranged from 75.41% to 78.66% (Table 3). These rRNA genes are located between *trnL1* and A + T-rich region and separated by *trnV*. There are no gaps between the rRNA genes. The total lengths of the 22 tRNA genes ranged from 1429 bp (*C. dentata*) to 1441 bp (*D. corrugata*), while the length of individual tRNA genes generally ranged from 60 to 70 bp, among which, eight tRNAs are encoded on the N-strand and the other 14 genes are encoded on the J-strand (Table 2). All tRNAs have the typical cloverleaf secondary structure, with the one exception being *trnS1*, in which the dihydrouridine (DHU) arm is absent and forms a simple loop (Figure 6), as seen in other insects [36,37,38,72,73]. Although the *trnS1* genes of these three species are almost the same size, with 67, 66, and 67 bp, respectively, there are significant differences in their structure. UCUs of them are located in the anticodon loops, which have been regarded as a molecular synapomorphy for Coleoptera by Sheffield et al. [74]. In addition, UG mismatches were detected in some tRNAs (Appendix A).

### 3.4. Non-Coding Region

Typically, the A + T-rich region also known as the control region (CR), is the largest non-coding region in mitogenome. The lengths of A + T-rich regions of these three species were 1614 bp (*C. copipes*), 1610 bp (*C. dentata*), 1668 bp (*D. corrugata*), respectively. The A + T-rich region in these three mitogenomes are located between *trnI* and *rrnS*. The A + T content (79.50–83.60%) of these three species was found to be higher than that of the whole mitogenome (71.76–76.59%), PCGs (69.70–75.23%), rRNAs (75.41–78.66%), and tRNAs (73.56–74.58%). Furthermore, compositional analysis showed that only the A + T-rich region of *D. corrugata* had a positive AT-skew among these three species.

In these three species, in addition to the large non-coding region, there are several small non-coding intergenic spacers, which are composed of less than 10 non-coding nucleotides in the mitochondria of most animals [75]. Nevertheless, in three species, longer than usual non-coding elements in the gene spacer region were found between the *trnS2* and *nad1* genes, with lengths of 18 in *C. copipes*, 24 in *C. dentata*, and 29 in *D. corrugata*. This spacer exists in many Coleoptera mitogenomes and may be used as a constant molecular marker of Coleoptera mitochondrial DNA [65].

### 3.5. Phylogenetic Analysis

A total of 18 buprestid species and two outgroups (*Heterocerus parallelus* and *Dryops ernesti*) were used for the phylogenetic relationship based on mitogenome data. In this study, phylogenetic trees utilizing three nucleotide sequence matrixes (13 PCGs, 2 rRNAs, and 2 rRNAs + 13 PCGs) from 20 species were constructed using different best-fit substitution models (Table 4). The results of phylogenetic analysis based on different datasets were almost identical. Although the ML and BI trees constructed from protein genes do not have the same topology, they do not differ much in their expression results (Figure 7 and Figure 8). Both ML and BI trees using two datasets (2 rRNAs and 2 rRNAs + 13 PCGs) produced identical topologies, with only slight differences in nodal support value (Figure 9 and Figure 10). For the same datasets, the node support values of BI trees are always higher than ML trees, which has often occurred in many previous studies of other taxa [76,77,78].

The results showed that the monophyly of Agrilinae is confirmed again, as all Agrilinae species formed to a single highly supported clade. Meanwhile, two outgroup taxa obviously separated from the clade of Buprestidae. Interestingly, the target species *C. copipes*, *C. dentata,* and *D. corrugata*, as well as two other Buprestinae species were clustered into a single branch with a high nodal support value. Moreover, the (*C. copipes* + *C. dentata*) + *D. corrugata*) clade and (*A. chinensis* + *M. acuminata*) clade displayed a sister group relationship; the *Chrysochroa fulgidissima* (Chrysochroinae) and *Acmaeodera* sp. (Polycestinae) formed a sister clade. Coomaniellini is more closely related to Dicercini than Melanophilini, which was in line with previous studies based on morphological data [79] and molecular data [15]. The genus *Coomaniella*, *Dicerca*, *Anthaxia,* and *Melanophila* were clustered together and form the Buprestinae branch based on the complete mitogenome data, however, the genus *Dicerca* belonged to Chrysochroinae in morphological classification. It is known that some scholars have supported the merger of Chrysochroinae and Buprestinae, indicating that there is no clear division between two subfamilies [12,13,14,15]. However, in this study, the *Dicerca* can be transferred from Chrysochroinae to Buprestinae based on the phylogenetic trees of this study, morphological data [79], and molecular data [15].

Although this study had contributed three new complete mitogenomes to the phylogeny of the Buprestidae, the interrelationships among the subfamilies and tribes still require more data to be determined completely. These questions will be well addressed in the future when sufficient numbers of complete mitogenomes of buprestid species are accumulated.

## 4. Conclusions

In this study, three mitogenomes (16,179–16,276 bp) were newly sequenced and annotated, which are the first complete mitogenome sequences to be reported in the tribes Coomaniellini and Dicercini. These three sequences have a positive AT-skew, in which, L2, I, S2, and F were the four most frequently used amino acids. Consistent with most studies on insects, only the secondary structure of *trnS1* is not a clover-leaf structure, but the absence of D-arm forms a simple loop. There is a 7 bp overlapping region between *atp8* and *atp6*, *nad4,* and *nad4l*, and the rearrangement phenomenon has not been detected in these three species. The gene *cox1* (Pi = 0.159) exhibited the lowest variability. The Ka/Ks value of *cox1* is the lowest, indicating that *cox1* gene has a relatively low evolutionary rate. The phylogenetic results showed that Coomaniellini is more closely related to Dicercini than Melanophilini, and the monophyly of Agrilinae is confirmed again. The authors recommend that the *Dicerca* should be transferred from Chrysochroinae to Buprestinae. However, the samples used in this study may be too limited for a more detailed analysis. Whether Dicercini can be completely transferred from Chrysochroinae to Buprestinae needs more species, especially those of Dicercini, to be further verified. The results of this study can provide new data for phylogenetic studies of the Buprestidae and improve our understanding of the characteristics of mitogenomic and the taxonomy of Buprestidae.

## Figures and Tables

**Figure 1 genes-13-01074-f001:**
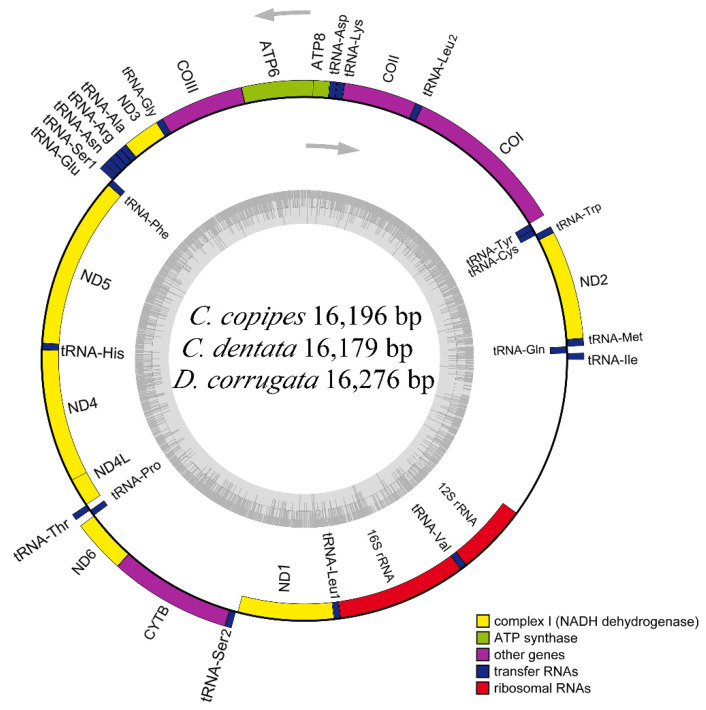
Gene maps of the mitogenomes of three Buprestidae species sequenced in this study.

**Figure 2 genes-13-01074-f002:**
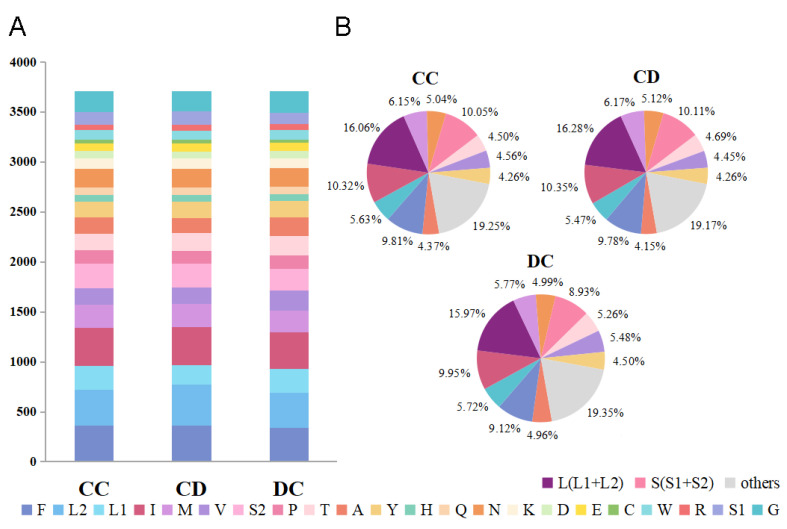
Numbers of different amino acids in the mitogenomes of the three Buprestidae species (**A**) and the percentage of top 11 amino acids with higher number (**B**); the stop codon is not included. CC: *Coomaniella copipes*, CD: *Coomaniella dentata*, DC: *Dicerca corrugata*.

**Figure 3 genes-13-01074-f003:**
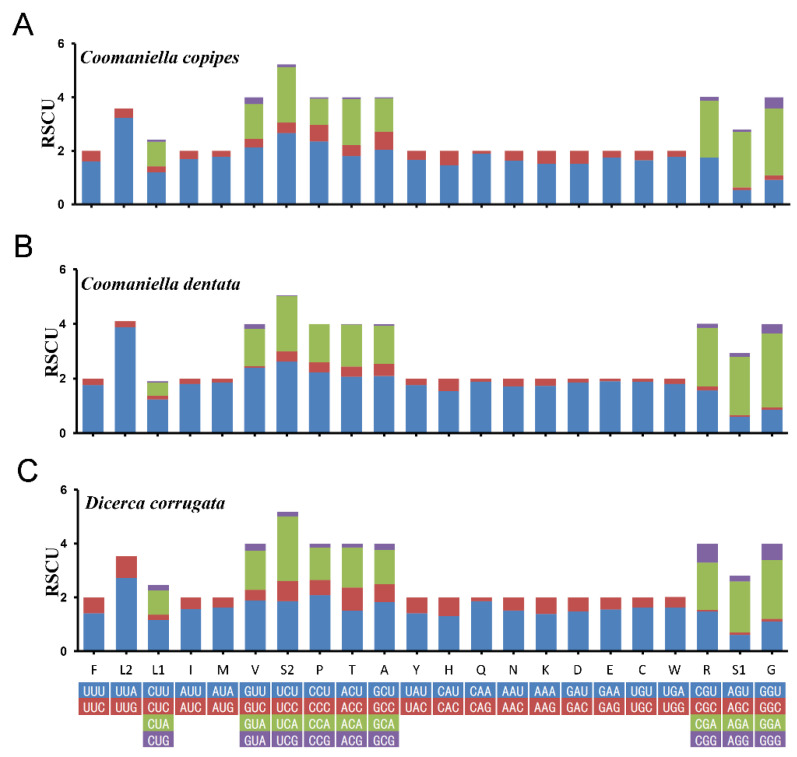
Relative synonymous codon usage (RSCU) of the mitogenomes of the three Buprestidae species; the stop codon is not included. *Coomaniella copipes* (**A**), *Coomaniella dentata* (**B**), *Dicerca corrugata* (**C**).

**Figure 4 genes-13-01074-f004:**
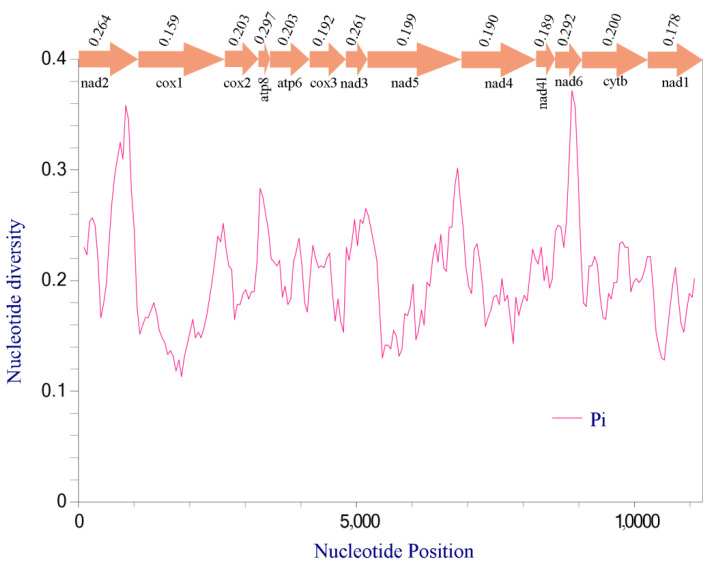
Nucleotide diversity (Pi) of 13 PCGs in three newly sequenced buprestid mitogenomes.

**Figure 5 genes-13-01074-f005:**
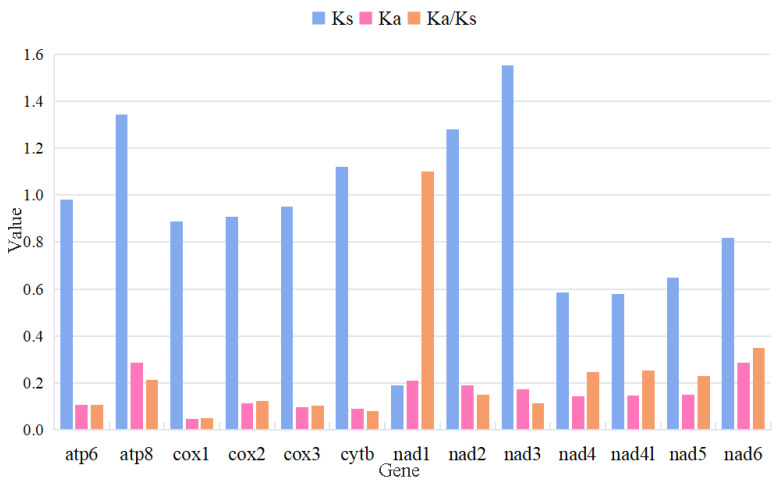
The ratio of Ka/Ks of 13 PCGs in three newly sequenced buprestid mitogenomes.

**Figure 6 genes-13-01074-f006:**
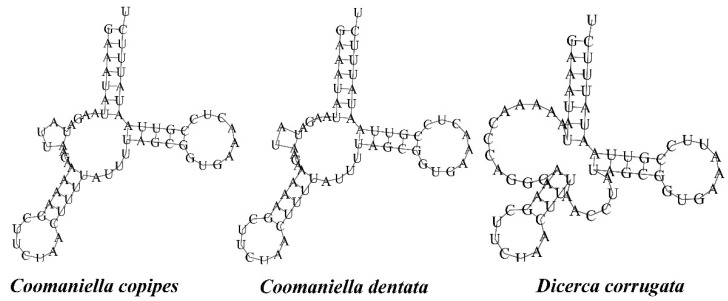
The *trnS1*-predicted secondary structures of tRNAs in the mitogenomes of the three buprestid species.

**Figure 7 genes-13-01074-f007:**
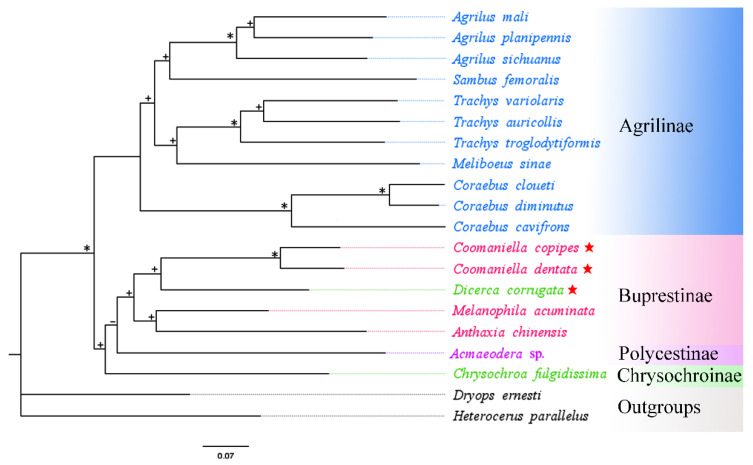
Phylogenetic relationships of 18 selected buprestid species using ML analyses based on 13 PCGs of mitogenomes. The symbols on the branches show bootstrap (ML tree). * ML bootstrap = 100; + ML bootstrap ≥ 50; − ML bootstrap < 50. Red star: means species which are newly sequenced in this study.

**Figure 8 genes-13-01074-f008:**
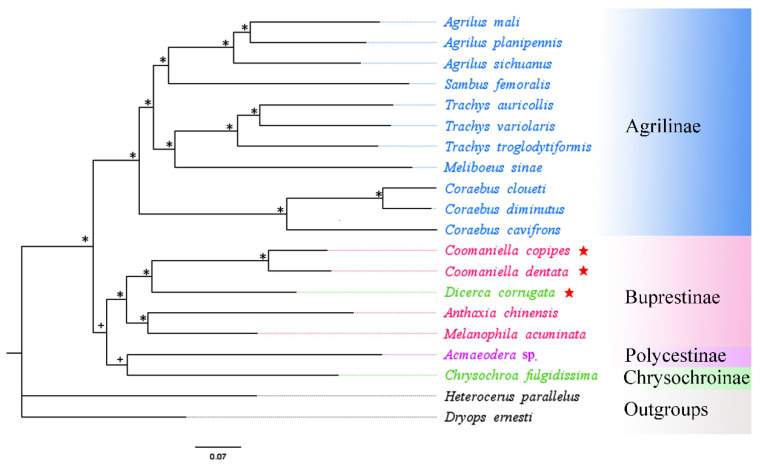
Phylogenetic relationships of 18 selected buprestid species using BI analyses based on 13 PCGs of mitogenomes. The symbols on branches show posterior probability (BI tree). * posterior probabilities = 1; + posterior probabilities ≥ 0.5; Red star: means species which are newly sequenced in this study.

**Figure 9 genes-13-01074-f009:**
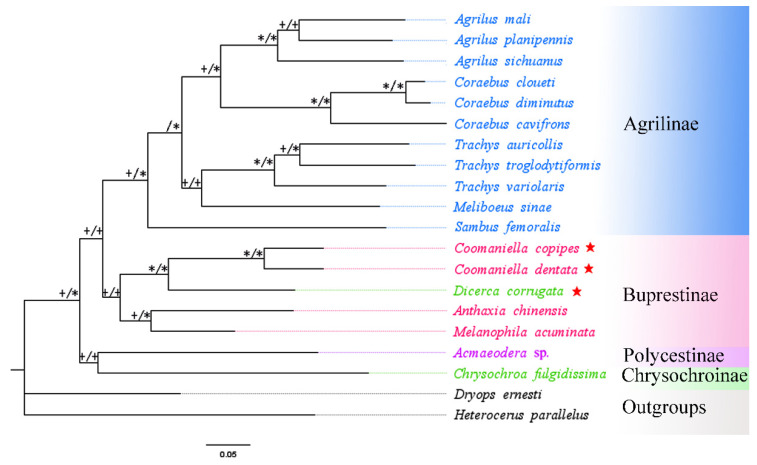
Phylogenetic relationships of 18 selected buprestid species inferred based on maximum-likelihood and Bayesian analyses of two rRNA genes. * ML bootstrap = 100 or posterior probabilities = 1; + ML bootstrap ≥ 50 or posterior probabilities ≥ 0.5; Red star: means species which are newly sequenced in this study.

**Figure 10 genes-13-01074-f010:**
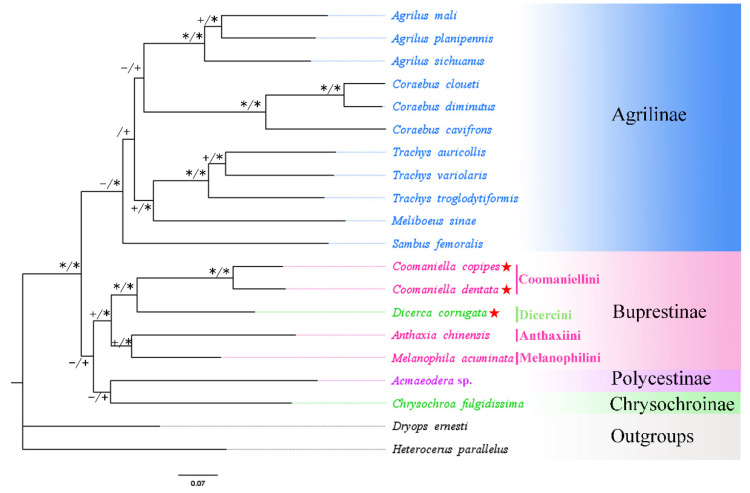
Phylogenetic relationships of 18 selected buprestid species inferred based on maximum-likelihood and Bayesian analyses of 13 protein-coding genes and two rRNA genes. * ML bootstrap = 100 or posterior probabilities = 1; + ML bootstrap ≥ 50 or posterior probabilities ≥ 0.5; − ML bootstrap < 50 or posterior probabilities < 0.5. Red star: means species which are newly sequenced in this study.

**Table 1 genes-13-01074-t001:** Taxonomy, GenBank accession numbers, and related information on the mitochondrial genomes used for the phylogenetic analysis.

No.	Family/Subfamily	Taxa	Accession No.	Genome Size (bp)	A + T%	AT-Skew	GC-Skew	Location/References
1	Buprestinae	*Melanophila acuminata*	MW287594	15,853	75.66	0.02	−0.25	[39]
2	Buprestinae	*Anthaxia chinensis*	MW929326	15,881	73.61	0.09	−0.29	[40]
3	Buprestinae	*Coomaniella copipes*	OL694145	16,196	74.47	0.03	−0.24	This study
4	Buprestinae	*Coomaniella dentata*	OL694144	16,179	76.59	0.01	−0.21	This study
5	Chrysochroinae	*Dicerca corrugata*	OL753086	16,276	71.76	0.09	−0.21	This study
6	Chrysochroinae	*Chrysochroa fulgidissima*	EU826485	15,592	69.92	0.15	−0.24	[36]
7	Agrilinae	*Coraebus diminutus*	OK189521	15,499	68.42	0.12	−0.25	[41]
8	Agrilinae	*Coraebus cloueti*	OK189520	15,514	69.27	0.11	−0.25	[41]
9	Agrilinae	*Meliboeus sinae*	OK189522	16,108	72.42	0.11	−0.22	[41]
10	Agrilinae	*Sambus femoralis*	OK349489	15,367	73.23	0.12	−0.18	[41]
11	Agrilinae	*Agrilus sichuanus*	OK189519	16,521	71.73	0.12	−0.21	[41]
12	Agrilinae	*Agrilus planipennis*	KT363854	15,942	71.90	0.12	−0.24	[42]
13	Agrilinae	*Agrilus mali*	MN894890	16,204	74.46	0.08	−0.18	[38]
14	Agrilinae	*Coraebus cavifrons*	MK913589	15,686	69.79	0.12	−0.18	[43]
15	Agrilinae	*Trachys auricollis*	MH638286	16,429	71.05	0.10	−0.20	[37]
16	Agrilinae	*Trachys troglodytiformis*	KX087357	16,316	74.62	0.10	−0.19	unpublished
17	Agrilinae	*Trachys variolaris*	MN178497	16,771	72.11	0.11	−0.21	[44]
18	Polycestinae	*Acmaeodera* sp.	FJ613420	16,217	68.41	0.11	−0.25	[45]
19	Heteroceridae	*Heterocerus parallelus*	KX087297	15,845	74.03	0.13	−0.24	unpublished
20	Dryopidae	*Dryops ernesti*	KX035147	15,672	72.98	0.07	−0.23	unpublished

**Table 2 genes-13-01074-t002:** The three newly annotated Buprestidae mitogenomes.

Gene	Strand	Position		Codons		Anticodon	IGN
		From	To	Start	Stop		
*nad2*	J	199/201/201	1221/1223/1223	ATT/ATT/ATT	TAA/TAA/TAA		−2/−2/−2
*cox1*	J	1405/1407/1405	2935/2937/2935	–/–/–	T/T/T		0/0/0
*cox2*	J	3001/3003/3001	3685/3684/3682	ATA/ATA/ATA	T/T/T		0/0/0
*atp8*	J	3822/3824/3817	3977/3979/3972	ATT/ATC/ATT	TAA/TAA/TAA		−7/−7/−7
*atp6*	J	3971/3973/3966	4645/4647/4640	ATG/ATG/ATG	TAA/TAA/TAA		−1/−1/−1
*cox3*	J	4645/4647/4640	5431/5433/5426	ATG/ATG/ATG	T/T/T		0/0/0
*nad3*	J	5494/5496/5490	5847/5849/5843	ATT/ATT/ATT	TAG/TAG/TAG		−2/−1/−2
*nad5*	N	6231/6229/6227	7950/7948/7946	ATT/ATT/ATT	T/T/T		0/0/0
*nad4*	N	8016/8013/8012	9351/9348/9347	ATG/ATG/ATG	T/T/T		−7/−7/−7
*nad4l*	N	9345/9342/9341	9629/9626/9631	ATG/ATG/ATG	TAA/TAA/TAA		2/2/2
*nad6*	J	9760/9757/9763	10,269/10,266/10,263	ATT/ATT/ATT	TAA/TAA/TAA		−1/−1/−1
*cytb*	J	10,269/10,266/10,263	11,411/11,408/11,405	ATG/ATG/ATG	TAG/TAG/TAG		−2/−2/−2
*nad1*	N	11,493/12,485/11,501	12,446/12,441/12,451	TTG/TTG/TTG	TAA/TAA/TAG		1/1/1
*trnI*	J	1/1/1	64/65/65			GAT	−3/−3/−3
*trnQ*	N	62/63/63	130/131/131			TTG	−1/0/0
*trnM*	J	130/132/132	198/200/200			CAT	0/0/0
*trnW*	J	1220/1222/1222	1284/1286/1287			TCA	−8/−8/−8
*trnC*	N	1277/1279/1280	1337/1339/1341			GCA	0/0/0
*trnY*	N	1338/1340/1342	1403/1405/1403			GTA	1/1/1
*trnL2*	J	2936/2938/2936	3000/3002/3000			TAA	0/0/0
*trnK*	J	3686/3685/3683	3755/3754/3752			CTT	0/0/0
*trnD*	J	3756/3755/3753	3820/3823/3816			GTC	1/0/0
*trnG*	J	5432/5434/5427	5493/5495/5489			TCC	0/0/0
*trnA*	J	5846/5849/5842	5908/5910/5904			TGC	−1/0/−1
*trnR*	J	5908/5911/5904	5971/5970/5970			TCG	−1/0/−3
*trnN*	J	5971/5971/5968	6037/6035/6032			GTT	0/0/0
*trnS1*	J	6038/6036/6033	6104/6101/6099			TCT	0/0/0
*trnE*	J	6105/6102/6100	6168/6165/6163			TTC	−1/−1/−1
*trnF*	N	6168/6165/6163	6230/6228/6226			GAA	0/0/0
*trnH*	N	7951/7949/7947	8015/8012/8011			GTG	0/0/0
*trnT*	J	9632/9629/9634	9694/9691/9696			TGT	0/0/0
*trnP*	N	9695/9692/9697	9758/9755/9761			TGG	1/1/1
*trnS2*	J	11,410/11,407/11,404	11,474/12,470/11,471			TGA	18/14/29
*trnL1*	N	12,448/12,443/12,453	12,510/12,505/12,517			TAG	0/0/0
*trnV*	N	13,800/13,789/13,800	13,867/13,856/13,869			TAC	0/0/0
*rrnL*	N	12,511/12,506/12,518	13,799/13,788/13,799				0/0/0
*rrnS*	N	13,868/13,857/13,870	14,582/14,569/14,608				0/0/0
A + T-rich region	J	14,583/14,570/14,609	16,196/16,179/16,276				0/0/0

The order of these three species in the table is as follows: *C. copipes*, *C. dentata,* and *Dicerca corrugata*. IGN = intergenic nucleotides; – not determined.

**Table 3 genes-13-01074-t003:** Summarized mitogenomic characteristics of the three buprestid species investigated in this study.

Species	PCGs			rRNAs			tRNAs			A + T-Rich Region		
	Size (bp)	A + T (%)	AT-Skew	Size (bp)	A + T (%)	AT- Skew	Size (bp)	A + T (%)	AT-Skew	Size (bp)	A + T (%)	AT-Skew
*C. copipes*	11,159	72.93	−0.16	2004	77.64	−0.03	1432	74.58	0.01	1614	81.16	−0.05
*C. dentata*	11,159	75.23	−0.16	1996	78.66	−0.01	1429	76.35	0.02	1610	83.60	−0.09
*D. corrugata*	11,150	69.70	−0.15	2021	75.41	−0.09	1441	73.56	0.02	1668	79.50	0.06

**Table 4 genes-13-01074-t004:** Best-fit models of three datasets used for phylogeny.

	ML	BI
13PCGs	GTR + F + I + G4	GTR + F + I + G4
2rRNAs	TVM + F + I + G4	GTR + F + I + G4
13PCGs + 2rRNAs	GTR + F + I + G4	GTR + F + I + G4

## Data Availability

The mitogenomes were deposited at NCBI, with accession numbers OL694145, OL694144, and OL753086.

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
