# Peer review of "First Report of Complete Mitochondrial Genome in the Tribes Coomaniellini and Dicercini (Coleoptera: Buprestidae) and Phylogenetic Implications"

_genes, 2022, doi:10.3390/genes13061074_

Round 1

Reviewer 1 Report

The paper "First report of complete mitochondrial genome in the tribes Coomaniellini and Dicercini (Coleoptera: Buprestidae) and phylogenetic implications" by Huang and colleagues is a beautiful scientific piece reporting for the first the mitochondrial genome sequences of these insect species. The authors report with great detail their findings, as well as a taxonomycal description of the investigated species (Table 1), a mitochondrial genome map of the sequenced species (Figure 1), detected changes in genomic sequence (Table 2), et cetera. The authors show an uncommon mastery of several topics, from sequencing to taxonomy, from secondary structure analysis (Figure 6) to genome annotation, from nucleotide profile statistics (Table 3) to phylogenetic analysis (Figure 10)

My only point is that the authors, throughput their study, investigate predominantly the nucleic acid sequences (genome and tRNAs), without discussing in detail variants in the protein sequences translated from the mitochondrially-encoded genes. Are these variants possibly associated to different protein functions (for example: substrate affinity, folding resistance to heat, ...)?

Minor points

- Paragraph 2.3: "Total 18" should be "A total of 18"
- "Conclusion" (page 13) should be paragraph 4, not 1.
- In the Conclusion paragraph, "the first complete mitogenome sequence" should be plural.

Author Response

Dear reviewer,please check the attachment.

Reviewer 2 Report

This is an important study. The authors have sequenced, assembled, and annotated three mitochondrial genomes (Coomaniella copipes, Coomaniella dentata, and Dicerca corrugata). I think this work is useful and has provided valuable information about phylogenetic analyses and evolutionary studies. I would suggest some necessary revisions as follows:

1- The abstract, institutional affiliation of authors, and keywords are missing.

2- In the introduction section, in the sentence "To date, 15 complete mitogenome sequences have been reported, details are shown in Table 1",  the authors mentioned 15 complete mitogenome sequences reported in table 1. However, there were 20 mitogenome sequences reported in the referred table. Please check this inconsistency.

3- Table 1 has many flaws that are required to be regenerated:

- The number/text has to fit in the cell.

- The serial numbers, after reaching 20, goes back to being 18.

- Accession numbers of 5 mitogenome sequences are not found in Genbank: OK189521, OK189520, OK189522, OK349489, and OK189519.

- The accession number of Dryopidae-Dryops ernesti is missing.

- The three Buprestinae-Coomaniella copipes, Buprestinae -Coomaniella dentata, Chrysochroinae Dicerca corrugate has just one accession number which is KX035147.

4- Table 2 has a few flaws that are required to be regenerated:

- To assist the reader in easily following the table, please split it into two tables: one for PCGs has a start and stop codon, and another for tRNAs and rRNAs that have no stop and start codon.

- It should be split into three columns for each species in the column "position from and to".

5- The figure and headings number should be in serial: 1, 2, 3, ..., so on. not 2, 3, 4, 5, 6, 1, 8, 9, 10.

6- Because there were five mitogenome sequences not found in Genbank, the phylogenetic trees must be re-analyzed without these mitogenomes allowing reproduce the trees.  

7- In the conclusions section, the authors should expand the conclusions to explain the contribution of having these newly sequenced mitogenomes to the scientific community.

Author Response

(The authors gave the same response as above.)

Round 2

Reviewer 2 Report

Thank you for addressing my comments. The current manuscript has much improved and will be of interest to the journal's readership.

Author Response

Dear reviewer,

Once again, thank you very much for your comments and suggestions.  Looking forward to our next cooperation!

Yours sincerely,

Xuyan Huang